# Molecular Quantification of Total and Toxigenic *Microcystis* Using Digital-Droplet-Polymerase-Chain-Reaction-Based Multiplex Assay

**DOI:** 10.3390/toxins17050242

**Published:** 2025-05-12

**Authors:** In-Su Kim, Hae-Kyung Park

**Affiliations:** 1Nakdong River Environment Research Center, National Institute of Environmental Research, Daegu 43008, Republic of Korea; factork@korea.kr; 2Department of Environmental Engineering, Chungbuk National University, Cheongju 28644, Republic of Korea

**Keywords:** toxigenic *Microcystis*, microcystins, digital droplet PCR, genus-specific primers, *mcyA*, *secA*, *Dolichospermum*, *Planktothrix*

## Abstract

The proliferation of harmful cyanobacteria, particularly *Microcystis*, poses significant risks to drinking and recreational water resources, especially under the influence of climate change. Conventional monitoring methods based on microscopy for harmful cyanobacteria management systems are limited in detecting toxigenic genotypes, hindering accurate risk assessment. In this study, we developed a digital droplet PCR (ddPCR)-based method for the simultaneous quantification of total and toxigenic *Microcystis* in freshwater environments. We targeted the *secA* gene, specific to the *Microcystis* genus, and the *mcyA* gene, associated with microcystin biosynthesis. Custom-designed primers and probes showed high specificity and sensitivity, enabling accurate detection without cross-reactivity. The multiplex ddPCR assay allowed for concurrent quantification of both targets in a single reaction, reducing the analysis time and cost. Application to field samples demonstrated good agreement with microscopic counts and revealed seasonal shifts in toxigenic genotype abundance. Notably, ddPCR detected *Microcystis* at very low densities—down to 7 cells/mL in the mixed cyanobacterial communities of field samples—even when microscopy failed, highlighting its utility for early bloom detection. This approach provides a reliable and efficient tool for monitoring *Microcystis* dynamics and assessing toxin production potential, offering significant advantages for the early warning and proactive management of harmful cyanobacterial blooms.

## 1. Introduction

The increase in population and subsequent human activities have led to land-use changes, urbanization, and expanded industrial and agricultural activities, consequently increasing pollutant loads in freshwater systems such as rivers, lakes, and reservoirs, thereby accelerating eutrophication [1,2]. Eutrophication inevitably enhances primary production and alters community structures, particularly triggering the proliferation of toxin-producing harmful cyanobacterial blooms (cyanoHABs), which pose serious threats to aquatic ecosystems, water resource utilization, and human health on a global scale [3,4,5,6,7,8]. Furthermore, ongoing global climate change is expected to increase the intensity, frequency, and duration of cyanoHAB events [9,10,11,12].

The major cyanobacterial genera responsible for cyanoHABs include *Microcystis*, *Aphanizomenon*, *Cylindrospermopsis*, *Dolichospermum*, and *Planktothrix* [7,13]. Among these, *Microcystis* is the most frequently reported bloom-forming cyanobacterium across diverse freshwater environments worldwide [4,14,15,16,17,18,19]. In the rivers and lakes of Korea, *Microcystis* blooms occur extensively during the warm summer months, significantly impairing water resource utilization [20,21]. *Microcystis* forms colonies with cells arranged irregularly, sparsely, or densely, within a common mucilage, which proliferate rapidly under warm conditions and often accumulate as surface scums [21,22].

The primary health risk associated with *Microcystis* blooms arises from the potent hepatotoxin they produce, known as microcystins. *Microcystis* is the most well-known producer of microcystins, although other cyanobacteria have also been reported to produce them [7,23,24]. Once absorbed into cells, microcystins inhibit protein phosphatases (PP1, PP2A, and PP5), leading to cytoskeletal destabilization and subsequent apoptosis and necrosis. High acute exposure can result in hepatic hemorrhage due to damage to sinusoidal capillaries, while chronic exposure at low doses (below 20 μg/kg body weight) may lead to phosphatase inhibition, abnormal cellular proliferation, hepatic hypertrophy, and tumor-promoting effects [25].

Microcystins are cyclic heptapeptides containing a unique ADDA moiety, and variations in amino acid residues result in over 250 known structural variants, such as microcystin-LR, -YR, and -RR [26]. Their biosynthesis involves a multi-enzyme complex, including polyketide synthases and non-ribosomal peptide synthetases, which enhance their chemical stability and influence their biodegradability in aquatic environments [27].

In 1998, the World Health Organization (WHO) established a provisional drinking water guideline value of 1 μg/L for microcystin-LR—the first regulatory measure targeting a cyanotoxin [28]. Many countries have since adopted or modified this guideline to ensure drinking water safety, implementing regular monitoring and management frameworks such as the Alert Level Framework (ALF) [7,29,30,31,32,33]. The ALF alert thresholds are typically based on cyanobacterial cell density, biovolume, or chlorophyll-a concentration, assessed using microscopy-based species identification and biomass measurements [30].

In addition to microscopic analysis, various monitoring technologies have recently been developed for the effective management of harmful cyanobacteria. Microcystins and other cyanotoxins are analyzed using liquid chromatography–tandem mass spectrometry (LC–MS/MS), with water samples collected through conventional sampling methods or using solid-phase adsorption toxin tracking (SPATT) samplers [34]. Satellite-based remote sensing techniques have also been developed to generate high-resolution chlorophyll-a and phycocyanin maps for bloom detection [35]. Additionally, machine learning models such as CyanoNet have demonstrated practical effectiveness in the detection and quantification of cyanobacteria by modeling cyanobacterial indices (CIs) from satellite data [36].

However, most existing monitoring techniques focus on measuring the total harmful cyanobacterial biomass and bloom expansion. Since not all cyanobacteria produce toxins, risk assessments based solely on total cyanobacterial biomass may overestimate public health risks associated with cyanotoxins [33,37,38]. The most accurate approach for evaluating toxin-associated risks involves directly analyzing toxin concentrations using LC–MS/MS or ELISA. However, these methods require substantial investments in high-cost analytical equipment and operation by highly trained personnel [33]. Furthermore, while direct toxin monitoring provides precise toxin concentration estimates, it does not account for non-specific health effects associated with cyanoHABs. Therefore, a time- and cost-effective tool is needed to simultaneously estimate the cyanobacterial biomass and toxin production potential for a more comprehensive risk assessment regarding cyanoHABs [7].

With advancements in PCR-based molecular techniques, studies have focused on detecting and quantifying genes responsible for the biosynthesis of toxins and odor compounds in specific cyanobacterial taxa [13,33,39,40,41,42,43,44]. Genetic methods primarily indicate the potential for toxin production rather than providing direct information on actual toxin concentrations. Nonetheless, integrating genetic methods into monitoring programs allows for the early warning of toxin-producing blooms and identification of toxin-producing taxa within mixed cyanobacterial populations. Additionally, genetic approaches facilitate high-throughput sample analysis [7].

Quantitative real-time PCR (qPCR) has been widely used for DNA quantification [42,45,46,47,48]; however, digital PCR (dPCR) has recently emerged as an alternative method [49,50,51,52]. Unlike qPCR, which estimates DNA quantities based on standard curves, but its precision and sensitivity decrease with low-concentration samples, dPCR enables absolute quantification without standard curves and maintains robustness in the presence of PCR inhibitors, offering greater accuracy and reproducibility [53,54].

In this study, we developed a genetic analysis method using Digital Droplet PCR (ddPCR), which offers superior accuracy, sensitivity, and reproducibility. This method was used to estimate the biomass of *Microcystis*, the most frequently occurring harmful cyanobacterium responsible for cyanoHABs worldwide, as well as the biomass of toxigenic *Microcystis*, which possess the microcystin synthetase gene and therefore have the potential to produce microcystins. The developed method was applied to field samples, and its applicability was evaluated by comparison with conventional microscopic analysis.

## 2. Results

### 2.1. Verification of secA and mcyA ddPCR Primers and Probes

The primers and probes were designed to specifically detect *Microcystis* spp. (Table 1). The *secA* gene, a component of the secYEG protein translocation channel in the cell membrane, was selected as the target for *Microcystis*-specific primer design [55]. To verify the genus-level specificity, both conventional PCR and ddPCR analyses were conducted using DNA from *Microcystis* and other cyanobacterial species. These included species of the same order (Chroococcales), such as *Aphanocapsa* sp. and *Synechococcus* sp., as well as species from different orders, including *Aphanizomenon flos-aquae*, *Dolichospermum planctonicum*, *Sphaerospermopsis aphanizomenoides*, *Cuspidothrix issatchenkoi*, *Cylindrospermopsis raciborskii*, and *Geitlerinema amphibium*. The results confirmed that the primers and probe reacted specifically with the genus *Microcystis* (Figure 1A). In contrast, the previously reported 16S rRNA gene primers and probe used in qPCR [56] yielded false-positive signals in non-*Microcystis* genera, demonstrating their lack of specificity for ddPCR analysis (Figure 1B).

Based on previously reported universal *mcyA* primers [57], we designed *Microcystis*-specific *mcyA* ddPCR primers and a probe set. While the universal *mcyA* primers amplified PCR products in all microcystin-producing cyanobacteria regardless of genus, the newly designed ddPCR primers and probe yielded positive signals exclusively in *Microcystis* strains that produce microcystins. No positive signals were observed in *Dolichospermum flos-aquae,* or *Planktothrix agardhii*, although these species produce microcystins, indicating the high specificity of the newly developed primers and probe for *Microcystis*-derived *mcyA* (Figure 1C).

### 2.2. Comparison of Sensitivity Between secA and mcyA Singleplex and Multiplex PCR Using ddPCR

To verify whether the designed primer and probe sets could be used simultaneously in a multiplex ddPCR analysis, both *secA* and *mcyA* genes were analyzed independently (singleplex) and concurrently (multiplex) using ddPCR in toxic *Microcystis* strains isolated from the Korean rivers. The *secA* probe was labeled with fluorescein amidite (FAM), and the *mcyA* probe was labeled with hexachlorofluorescein (HEX), enabling dual-channel detection. In singleplex ddPCR analysis of four toxic *Microcystis* strains (*M. flos-aquae* [NRERC-205], *M. viridis* [NRERC-224], *M. novacekii* [NRERC-225], and *M. aeruginosa* [NRERC-230]), *secA* gene copy numbers ranged from 3.9 to 7.0 copies/μL (mean: 5.5 copies/μL), while *mcyA* gene copy numbers ranged from 4.4 to 7.5 copies/μL (mean: 5.9 copies/μL), with variation among strains. In multiplex ddPCR, *secA* gene copy numbers ranged from 3.9 to 7.4 copies/μL (mean: 5.6 copies/μL), while *mcyA* gene values ranged from 4.2 to 8.0 copies/μL (mean: 5.9 copies/μL). The results of the multiplex PCR were almost identical to those of the singleplex PCR, indicating that the two primer and probe sets for *secA* and *mcyA* did not interfere with each other in multiplex PCR and could be stably measured (Figure 2A).

To assess field applicability, water samples with low and high *Microcystis* cell densities were selected from CH site in 2022 based on microscopic counts. The low-density sample (860 cells/mL) was collected on 23 May 2022 and the high-density sample (131,000 cells/mL) on 8 August 2022. These samples were diluted to below the detection threshold and analyzed using both singleplex and multiplex ddPCR, as with the cultured strains. In the singleplex PCR analysis, the *secA* and *mcyA* gene copies in the low-density sample were 1.8–2.4 and 0.6–0.8 copies/μL, with averages of 2.1 and 0.7 copies/μL, respectively. In the high-density sample, the *secA* and *mcyA* gene numbers were 5.0–5.4 and 3.4–3.5 copies/μL, with averages of 5.4 and 3.4 copies/μL, respectively. Similar values were obtained from the multiplex ddPCR analysis: the *secA* and *mcyA* gene copy numbers were 1.7–2.3 and 0.6–0.9 copies/μL (mean: 2.1 and 0.7 copies/μL), in the low-density sample and 5.3–5.5 and 3.3–3.5 copies/μL (mean: 5.4 and 3.4 copies/μL) in the high-density sample, respectively. These results further confirmed that both primer/probe sets were reliable and compatible for multiplex ddPCR in both cultured and environmental samples (Figure 2B).

### 2.3. Copy Number per Cell of secA and mcyA Genes in Microcystis Cells

The copy number of the *secA* gene per cell was determined using sub-cultured *Microcystis* strains isolated from the Nakdong River. The values ranged from 1.1 to 16.4 copies/cell, with an average of 7.1 copies/cell. Average copy numbers per cell by strain were as follows: *M. aeruginosa,* 7.8; *M. flos-aquae,* 2.0; *M. ichthyoblabe*, 2.2; *M. wesenbergii*, 14.1; *M. viridis,* 7.3; and *M. novacekii,* 8.7 copies/cell. These results showed differences between species, with *M. wesenbergii* exhibiting significantly higher copy numbers (10.4–16.5 copies/cell) compared to other species. Variation was also observed among strains of the same species—for example, *M. aeruginosa* NRERC-221 had an average of 4.0 copies/cell, whereas NRERC-230 had 10.3 copies/cell. In contrast, *M. viridis* NRERC-224 and NRERC-227 showed similar copy numbers (8.0 and 7.1 copies/cell, respectively). In mixed field populations of *Microcystis* collected from the Nakdong River basin, the *secA* gene copy number per cell was more consistent, ranging from 1.1 to 4.8 copies/cell, with an average of 2.8 copies/cell (Figure 3A).

Among toxigenic *Microcystis* strains, the average copy number per cell of the *secA* gene was 6.3 copies/cell, while the *mcyA* gene copy number averaged 6.3 copies/cell (ranging 1.1–11.6 copies/cell). Although variability was observed among strains, *secA* and *mcyA* gene copy numbers per cell were identical within each toxigenic strain (Figure 3B). Since toxigenic and non-toxigenic cells were mixed in the mixed field populations of *Microcystis* from the Nakdong River, it was not possible to separately count the cell number of toxigenic cells and, thus, the *mcyA* gene copy number per cell could not be calculated.

### 2.4. Comparison Between Microscopic Analysis and Genetic Quantification of Field Samples

From May to December 2022, water samples were collected twice a month from CH site in the Nakdong River, and *Microcystis* cell densities were measured microscopically. The cell densities ranged from 465 to 131,000 cells/mL. *Microcystis* first appeared in late May with rising water temperatures, followed by a significant increase in late June. Cell density slightly decreased during the rainy season but reached its peak in early August. After mid-August, heavy rainfall and typhoons led to a rapid decline that persisted through December. Regarding the dominance of species within the *Microcystis* genus, *M. aeruginosa* was the dominant species for most of the monitoring period; however, *M. wesenbergii* was dominant on 19 July and 7 November, while *M. viridis* was subdominant (>30%) on 7 June and 4 July (Figure 4A and Table 2).

Simultaneously, *secA* and *mcyA* gene copy numbers were measured using ddPCR, and converted to cell densities to estimate total *Microcystis* (*secA* gene) and toxigenic *Microcystis* (*mcyA* gene) cell densities. For this, a conversion factor of 2.8 copies/cell—derived from the average *secA* gene copy number per cell (2.8) in mixed field populations of *Microcystis* collected from the Nakdong River—was applied. The estimated total *Microcystis* cell density based on the *secA* gene copy number ranged from 136 to 196,198 cells/mL. Comparison between genetic and microscopic analyses yielded a coefficient of variation ranging from 7.0 to 102.4% (mean: 37.0%). For samples exceeding 10,000 cells/mL, the variation was lower, ranging from 16.4 to 47.3% (mean: 26.5%). Nonetheless, the temporal trends in cell density were consistent between methods, that is, the increase period and decrease period of the ddPCR estimate corresponded with the microscopic count. Additionally, the *secA* gene was detected in the May 9 sample—prior to visible detection by microscopy—at an estimated 136 cells/mL (Figure 4A).

Because the field samples contained a mixture of toxigenic and non-toxigenic *Microcystis*, the *mcyA* copy number per toxigenic cell could not be independently determined. Therefore, the same 2.8 copies/cell conversion factor was applied to *mcyA*, based on the 1:1 ratio of *secA* versus *mcyA* genes observed in cultured toxigenic strains (Figure 3B). The proportion of toxigenic *Microcystis* cells ranged from 5.2% (7 cells/mL) to 76.9% (29,066 cells/mL), with an average of 38.3%. Among the 15 samples, toxigenic *Microcystis* were dominant (>50%) in only four samples (27%), while non-toxigenic *Microcystis* predominated in the rest. The relative abundance of toxigenic *Microcystis* was relatively high during the summer bloom period (60.0% on 7 June, 76.9% on 4 July, 63.0% on 8 August, and 51.3% on 22 August), but declined in autumn following heavy rains, causing a sharp decline in the total cell density (Figure 4B).

## 3. Discussion

In this study, we developed a ddPCR-based gene quantification method to estimate the biomass of total *Microcystis* as well as toxigenic *Microcystis* genotypes. Previous studies have proposed several gene targets for the quantification of total *Microcystis*, including 16S rRNA, 16S–23S ITS, and *cpcBA* [47,58]. However, when we applied the primers and probe targeting 16S rRNA—commonly used in quantitative PCR (qPCR) analyses—to ddPCR, we observed non-specific amplification in other cyanobacterial genera. This result underscored the need for a new gene target capable of specifically detecting *Microcystis*. After reviewing the NCBI database and relevant studies, we selected the *secA* gene, which encodes a key component of the protein translocation system and regulates the function of the *secYEG* channel through ATPase activity. The *secA* gene has been employed as a supplementary marker for detecting and classifying pathogenic bacteria such as *Phytoplasma* [59,60,61]. The primers and probes that we designed for *secA* exhibited high sensitivity and specificity, reacting exclusively with *Microcystis* while eliminating the non-specific amplification observed with 16S rRNA primers (Figure 1B). These results suggest that the *secA* is a reliable genetic marker for detecting and quantifying *Microcystis* within mixed cyanobacterial communities.

Meanwhile, microcystins are known to be produced not only by *Microcystis* but also by other cyanobacterial genera such as *Dolichospermum*, *Aphanizomenon*, and *Planktothrix* [7]. To assess microcystin production, previous studies have widely targeted genes involved in ADDA synthesis, such as *mcyE*, as well as other genes within the microcystin biosynthesis cluster [26,43,62,63]. In this study, we selected *mcyA*, which is located in a different region of the open reading frame and is involved in the structural variation in microcystin residues [27]. The primers and probe that we designed for *mcyA* produced positive signals in all toxigenic *Microcystis* strains that were confirmed to produce microcystins by ELISA analysis, while no signals were observed in non-toxigenic strains. Furthermore, no amplification was observed in toxigenic *Dolichospermum* and *Planktothrix* strains, confirming the specificity of *mcyA*-based detection for *Microcystis* (Figure 1C). Thus, the *mcyA* primers/probe set developed in this study can be effectively used to estimate the biomass of toxigenic *Microcystis* and identify microcystin-producing genera within complex cyanobacterial communities. However, detecting a single gene, such as *mcyA* or *mcyE*, may not be sufficient to definitively confirm microcystin production. Future studies should consider the inclusion of additional genes in the microcystin biosynthesis cluster, such as *mcyB*, *mcyC*, and *mcyD*, to improve the accuracy of toxin production assessments.

The ddPCR has gained attention as a novel and precise method for detecting and quantifying cyanobacteria, providing higher accuracy and reproducibility than qPCR [40,48,54,64,65,66,67]. Zhang et al. (2014) [62] used qPCR to analyze the abundance of *Microcystis* spp. and *Cylindrospermopsis raciborskii* in aquatic environments, reporting a detection limit of 1 × 10^3^ cells/mL for *Microcystis* spp. While their qPCR-based quantification results aligned well with microscopic observations, they noted limited sensitivity of qPCR for low-concentration samples. Similarly, Te et al. (2015) [40] compared qPCR and ddPCR, concluding that ddPCR was more suitable for environmental sample analysis. According to their findings, the limit of detection (LOD) for *Microcystis* spp. was 8.5 × 10^6^ copies/reaction in qPCR, whereas ddPCR achieved a significantly lower LOD of 3.52 copies/reaction. The ddPCR also exhibited greater resistance to background noise from other species, allowing accurate quantification even at low target gene concentrations. Moreover, ddPCR is well-suited for multiplex PCR applications, enabling the simultaneous detection of multiple target genes within a single sample. This feature reduces the analysis time and cost while maintaining accuracy, making ddPCR an efficient tool for cyanobacterial studies [65,66,67]. In this study, we developed a multiplex ddPCR assay that simultaneously quantifies total and toxigenic *Microcystis*, thereby enabling rapid biomass estimation of both toxigenic and non-toxigenic genotypes.

Notably, we observed significant variations in the *secA* gene copy number per cell across different *Microcystis* strains. Similar variability has been reported in *Sphaerospermopsis* quantification using the *rbcLX* gene [41] and other cyanobacterial gene copy number analyses based on qPCR [68,69]. These differences were also observed in the environmental samples, suggesting that gene copy number may reflect the strain composition in natural populations influenced by environmental conditions such as nutrient availability and light intensity. This variability may reflect adaptive strategies, where specific strains dominate under certain environmental conditions. Previous studies have also noted such variations in gene copy numbers among both freshwater and marine cyanobacteria [68,69,70,71,72,73]. For example, Griese et al. (2011) demonstrated the growth-stage-dependent regulation of gene copy numbers in *Synechocystis* sp. PCC6803 [74], while Riaz et al. (2021) reported the precise regulation of gene copy numbers in *Thermosynechococcus elongatus* E542 in response to growth stage and nutrient availability [69]. These findings indicate that cyanobacteria can exhibit monoploid, oligoploid, or polyploid states as part of their response to external stressors [73,74]. Notably, the variability in gene copy number per cell in mixed *Microcystis* populations was smaller than that observed among individual strains. This suggests that while the gene copy number variability may limit the accuracy of estimating cell counts from gene copy numbers, such limitations could be overcome through further research involving diverse environmental samples.

Interestingly, the copy numbers of *secA* and *mcyA* remained consistent across toxigenic *Microcystis* genotypes, regardless of strain-specific variation. This may imply a strong regulatory linkage between these genes, possibly reflecting a balance between cellular metabolism and toxin biosynthesis. Since *secA* is involved in protein translocation and *mcyA* in microcystin biosynthesis, coordinated copy number regulation may indicate an intrinsic connection. However, this hypothesis remains speculative, as no other studies to date have explicitly confirmed this relationship. Further investigation is needed to explore strain-specific gene expression patterns and adaptive mechanisms.

When we estimated the *Microcystis* cell density from the gene analysis results of field samples collected from the CH station using the average *secA* gene copy number per cell obtained from mixed *Microcystis* populations from the Nakdong River system, the results mirrored seasonal trends observed in microscopic counts, although the absolute values differed. This discrepancy may arise from temporal variations in *Microcystis* species composition, growth stages, and environmental factors affecting gene copy numbers per cell. Nevertheless, the trend similarity suggests that genus-specific genetic quantification analysis holds great potential for estimating *Microcystis* biomass.

Furthermore, using the *mcyA* gene copy number measured by ddPCR, we estimated that toxigenic *Microcystis* comprised approximately 38% of the total *Microcystis* population. This implies that health risks from toxins during *Microcystis* blooms in the field may be significantly lower than what would be estimated based on microscopic analysis alone. Additionally, the proportion of toxigenic *Microcystis* increased during the warmer summer months, suggesting that the developed genetic method is well-suited for investigating environmental factors influencing toxigenic genotype dominance.

Importantly, *secA* gene detection in spring samples—where microscopic analysis failed to identify *Microcystis*—demonstrates that ddPCR can detect *Microcystis* even at extremely low cell densities during early bloom stages. This is crucial for managing toxic bloom development. Unlike filamentous cyanobacteria, which form filaments of a few to dozens of cells, *Microcystis* typically forms colonies of several to hundreds of cells. This has significant implications for managing toxic blooms, as *Microcystis* often exists in less than one colony of a few to several dozen cells per milliliter during the early bloom stages, making early-stage detection challenging via microscopy. The ability to detect toxigenic genotypes at trace levels well before bloom development is crucial for timely risk assessment and water quality management.

The production of microcystins plays a key role in *Microcystis* survival and environmental adaptation, with toxin gene expression being regulated by temperature, nutrient availability, pH, and light intensity [75,76,77,78]. Studies have shown that *Microcystis* growth rates and microcystin synthesis increase at higher temperatures [75], while high phosphorus levels favor toxigenic *Microcystis* dominance [76]. Additionally, alkaline pH (8–9) enhances toxin gene expression [77], and UV stress induces microcystin biosynthesis [78]. Given these findings, climate change and eutrophication may prolong toxigenic genotype dominance, further emphasizing the need for proactive management strategies.

The World Health Organization (WHO) recommends the Alert Levels Framework (ALF) for managing cyanobacterial risks in drinking and recreational water [7,32]. Conventional microscopic counting is time consuming, labor intensive, highly dependent on analyst expertise, and unable to distinguish between toxigenic and non-toxigenic strains. In contrast, the ddPCR-based approach developed in this study enables the rapid, sensitive detection of *Microcystis*, even at low concentrations, making it highly advantageous for early warning systems. Furthermore, the simultaneous analysis of *secA* and *mcyA* genes enables estimation of the proportion of toxigenic genotypes within *Microcystis* populations, providing valuable insights for cyanobacterial toxin management.

## 4. Conclusions

In this study, we developed a ddPCR-based genetic method for the simultaneous quantification of total and toxigenic *Microcystis* in freshwater environments. Targeting the *secA* gene, which is specific to the *Microcystis* genus and the *mcyA* gene, associated with microcystin biosynthesis, we demonstrated the method’s high specificity and sensitivity through both laboratory validation and field application.

The developed multiplex ddPCR assay successfully quantified both *secA* and *mcyA* gene copy numbers in environmental samples without cross-interference between the target genes. This method allows for the simultaneous detection and quantification of total and toxic *Microcystis*, which can produce microcystins. Estimated cell densities derived from gene copy numbers were consistent with trends observed in microscopic counts, while ddPCR exhibited superior sensitivity, detecting *Microcystis* even at low cell densities undetectable by microscopy.

This ddPCR-based method provides a sensitive and efficient tool for monitoring *Microcystis* blooms and distinguishing between toxigenic and non-toxigenic populations. This method can complement existing monitoring technologies such as microscopy and LC–MS/MS, offering a faster and more efficient approach for providing an early warning, managing water resources, and mitigating the risks posed by *Microcystis*.

## 5. Materials and Methods

### 5.1. Cyanobacterial Strains and Culture Conditions

To develop *Microcystis*-specific gene primers, six species and ten strains of *Microcystis—Microcystis flos-aquae*, *M. ichthyoblabe*, *M. aeruginosa*, *M. wesenbergii*, *M*. *viridis*, and *M. novacekii*—were used as positive controls. All strains were isolated from Korean rivers and lakes and maintained through sub-culturing. Among them, six strains were toxigenic (microcystin-producing) and four were non-toxigenic (Table 3).

As negative controls, strains from the orders *Nostocales* and Oscillatoriales—including *Aphanizomenon flos-aquae*, *Cylindrospermopsis raciborskii*, *Dolichospermum planctonicum*, *Sphaerospermopsis aphanizomenoides*, *Cuspidothrix issatschenkoi*, and *Geitlerinema amphimium*—were also isolated from the Nakdong River and sub-cultured. Additionally, strains from the order Chroococcales order (*Aphanocapsa* sp. [AG10016] and *Synechococcus* sp. [AG20470]) were obtained from the Korean Collection for Type Cultures (KCTC).

To verify the *Microcystis* specificity of the *mcyA* gene primers, toxigenic strains of *Dolichospermum flos-aquae* (NIVA-CYA 656) and *Planktothrix agardhii* (NIVA-CYA 855) were obtained from the Norwegian Culture Collection of Algae (NORCCA, Oslo, Norway). *Microcystis* strains possessing the microcystin synthesis gene *mcyA* were classified as toxigenic *Microcystis*. All cyanobacterial strains were cultured in CB medium [79] under controlled conditions: 20 °C, light intensity of 40 μEm^−2^s^−1^, and 14:10 h light/dark cycle.

### 5.2. Field Sample Collection and Pretreatment

To calculate the gene copy number per *Microcystis* cell in natural water bodies, *Microcystis* colonies were collected from three lake sites (Yeongju Lake [YJ], Sannam Reservoir [SN], and Junam Reservoir [JN]) and eight river sites (Nakdan [ND], Gumi [GM], Chilgok [CG], Changnyeong-Haman [CH], ND1, ND2, ND3, and ND4) along the Nakdong River, a large river flowing toward southern coast of the Republic of Korea [21], between July and August 2024 (Figure 5).

Additionally, to apply the developed ddPCR method to field samples, surface water samples (2 L) were collected twice monthly from May to December 2022 at the CH site of the Nakdong River. Samples were transported to the laboratory at 4 °C. For microscopic analysis, a portion of each sample was transferred to a glass bottle and preserved with Lugol’s iodine solution (final concentration: 0.3%). For genetic analysis, 50 mL of the sample was filtered through a 0.45 μm MicronSep nitrocellulose membrane disk (GVS Life Sciences, Findley, OH, USA) and stored at −80 °C until use.

*Microcystis* cell density was determined by examining 1 mL of the fixed sample using a Sedgwick-Rafter counting chamber under a phase-contrast microscope (ZEISS, Oberkochen, BW, Germany) at 100–400× magnification. Species identification was based on colony morphology, following references [22] and [80]. Colonies were disrupted using an ultrasonic processor (Sonics Vibra Cell, Sonics and Materials Inc., VCX 750, Newtown, CT, USA) [81] and the number of cells of the resulting cell suspension were counted to calculate cell density (cells/mL).

### 5.3. DNA Extraction

For genomic DNA extraction, both cultured and environmental samples were filtered through a 0.45 μm MicronSep nitrocellulose membrane disk (GVS Life Sciences, Findley, OH, USA) and stored at −80 °C until use. Genomic DNA was extracted using the DNeasy PowerBiofilm Kit (QIAGEN, Hilden, Germany), following the manufacturer’s instructions. Frozen filters were thawed at room temperature for five minutes prior to extraction. The concentration and purity of the extracted genomic DNA was measured using an Infinite M200 PRO instrument. Extracted DNA was stored at −20 °C until analysis.

### 5.4. Primer Design and Validation

To develop *Microcystis*-specific primers, *secA* gene sequences from major harmful cyanobacteria—including *Aphanizomenon*, *Dolichospermum*, *Cuspidothrix*, and *Cylindrospermopsis*—as well as from morphologically similar taxa like *Aphanocapsa* and *Synechococcus*, were obtained from NCBI GenBank. The collected sequences were aligned using MEGA 6.0 to identify regions unique to *Microcystis*.

For detection of the *mcyA* gene, which is involved in microcystin biosynthesis, primers were designed based on sequences from previously validated primer set, ensuring specificity to the *Microcystis mcyA* gene.

Primers and the probe for ddPCR were designed in accordance with the Bio-Rad Digital Droplet PCR Applications Guide. Specifically, primers were selected to have a GC content of 50–60% and a melting temperature (Tm) between 50 and 65 °C. The probe was designed to have a GC content of 30–80% and a Tm 3–10 °C higher than that of the primers. The amplicon length was maintained at under 250 bp.

The *Microcystis*-specific primers and probe were validated by comparing the results with those obtained using 16S rRNA-specific primers from previous studies [19], and the *Microcystis mcyA*-specific primers and probe were compared with analyses of microcystin-producing strains from other genera.

### 5.5. The ddPCR Analysis

For ddPCR, each reaction mixture was prepared in a final volume of 20 μL comprising 10 μL of ddPCR^TM^ Supermix for Probe (no dUTP) (Bio-Rad Laboratories Inc., Munich, Germany), 0.9 μL each of 10 pmol forward and reverse primers, 0.25 μL of a 10 pmol probe, and 1 μL of the sample DNA, with sterile distilled water used to adjust the final volume. The prepared reaction mixtures were loaded into a 96-well plate and partitioned into approximately 20,000 droplets of ~1 nanoliter each using the Bio-Rad QX200^TM^ Droplet Generator (Bio-Rad Lab Inc., Hercules, CA, USA). The generated droplets were then transferred to a new 96-well plate and subjected to PCR in a T100 Thermal Cycler (Bio-Rad Lab Inc., Hercules, CA, USA). The PCR protocol consisted of an initial denaturation at 95 °C for 10 min, followed by 40 cycles of 95 °C for 30 s and 56 °C for 40 s, with a final hold at 4 °C to complete the reaction.

After amplification, the plate was moved to a Droplet Reader (Bio-Rad Lab Inc., Hercules, CA, USA) to quantify the droplets by measuring the FAM or HEX fluorescence signals. Signal analysis was carried out using QuantaSoft^TM^ software version 1.7.4 (Bio-Rad Laboratories Inc., Hercules, CA, USA). To ensure data reliability, only results with a total droplet count of at least 12,000 were included in the analysis, as recommended by the Bio-Rad Digital Droplet PCR Applications Guide. All ddPCR-related reagents were obtained from Bio-Rad (Bio-Rad Laboratories Inc., USA).

### 5.6. Calculation of Microcystis Gene Copy Number per Cell Using ddPCR Analysis and Field Application

To calculate the gene copy number per *Microcystis* cell, cell densities of both cultured monoculture *Microcystis* strains and a mixed field population were enumerated. Samples containing 1000–20,000 cells were filtered through a 0.45 µm MicronSep nitrocellulose membrane disk (GVS Life Sciences, Findley, OH, USA). The filtered samples underwent genomic DNA extraction followed by ddPCR analysis targeting the gene of interest. Each sample was analyzed in six replicates.

The copy number per cell was calculated using the following formula:$$CN=\frac{PS\times DE}{CD\times V}$$

*CN*: Gene copy number per cell (copies/cell)

*PS*: Positive signal per well (copies/μL)

*DE*: DNA elution volume (μL)

*CD*: Counted cell density (cells/mL)

*V*: Filtered volume (mL)

To apply this method to field samples from the CH site (2022), the *secA* gene and *mcyA* gene copy numbers obtained via dd PCR were used to estimate the cell density of total and toxigenic *Microcystis,* respectively. These estimated cell densities were then compared with those obtained via microscopic enumeration. The conversion to cell density was calculated as follows:$$ECD=\frac{PS\times DE\times DM}{ACN\times V}$$

*ECD*: Estimated cell density (cells/mL)

*PS*: Positive signal per well (copies/μL)

*DE*: DNA elution volume (μL)

*DM*: Dilution multiple

*ACN*: Average gene copy number per cell (copies/cell)

*V*: Filtered volume (mL)

In this study, an average *secA* gene copy number of 2.8 copies per cell, derived from mixed field populations of *Microcystis* collected from the Nakdong River, was used to convert both the *secA* and *mcyA* gene copy numbers obtained using ddPCR into cell densities.

### 5.7. Analysis of Microcystins

For the analysis of microcystins, cultured *Microcystis* samples were placed in 50 mL conical tubes and disrupted using an ultrasonic homogenizer (Sonics Vibra Cell, Sonics and Materials Inc., VCX 750, Newtown, CT, USA). The resulting cell lysate was filtered through 0.45 μm syringe filter and then transferred into a clean tube. Microcystin concentrations were measured using a Microcystins (Adda specific) ELISA kit (Enzo Life Sciences, Farmingdale, NY, USA), following the manufacturer’s instructions.

### 5.8. Statistical Analysis

All data are presented as the mean ± standard error of the mean (SEM). Differences between samples were assessed using an independent two-sample *t*-test (two-tailed). A *p*-value < 0.05 was considered statistically significant.

## Figures and Tables

**Figure 1 toxins-17-00242-f001:**
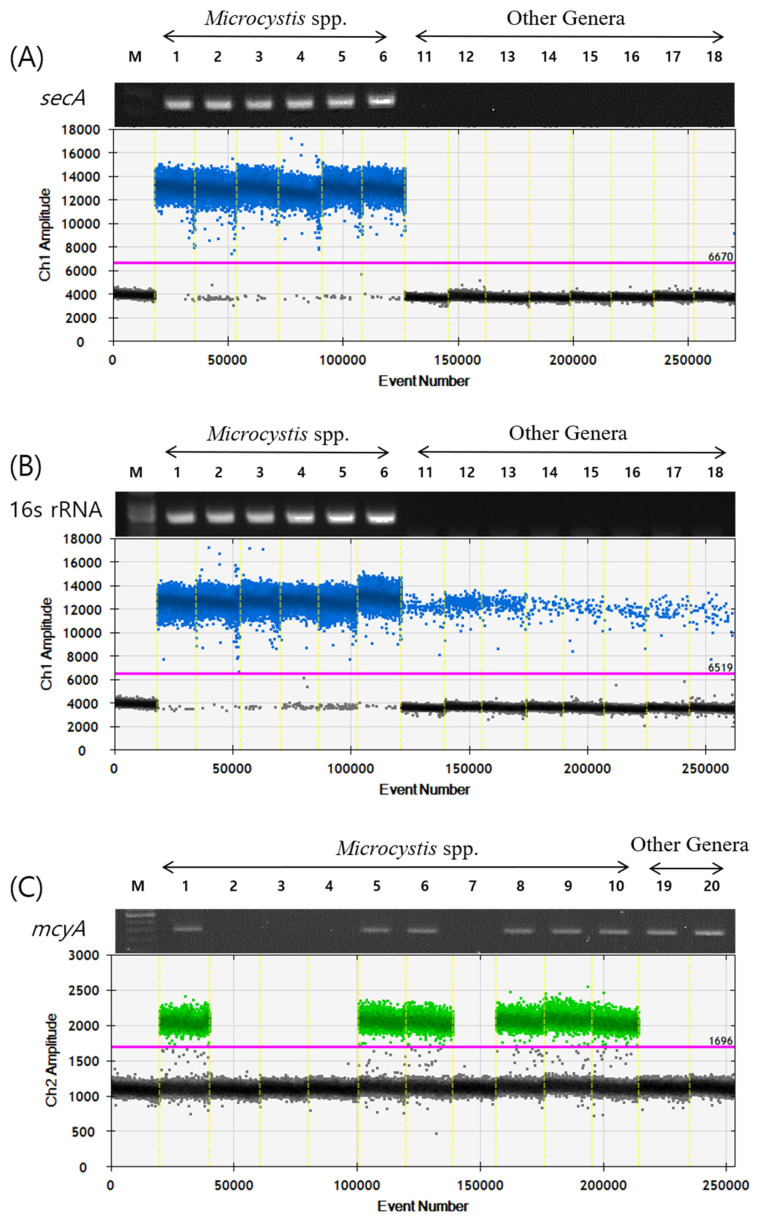
Detection of target genes in *Microcystis* spp. and other genera using PCR and ddPCR. The upper panel displays gel electrophoresis results showing amplification band, and lower panel shows droplet fluorescence data. (**A**) *secA* gene; (**B**) 16s rRNA gene; (**C**) *mcyA* gene; Droplets with fluorescence intensity above the threshold (pink line) are considered positive, indicating the presence of target DNA, while those below are considered negative. M: SiZer-100 DNA marker; event number: total droplets; numbers: refer to Table 3.

**Figure 2 toxins-17-00242-f002:**
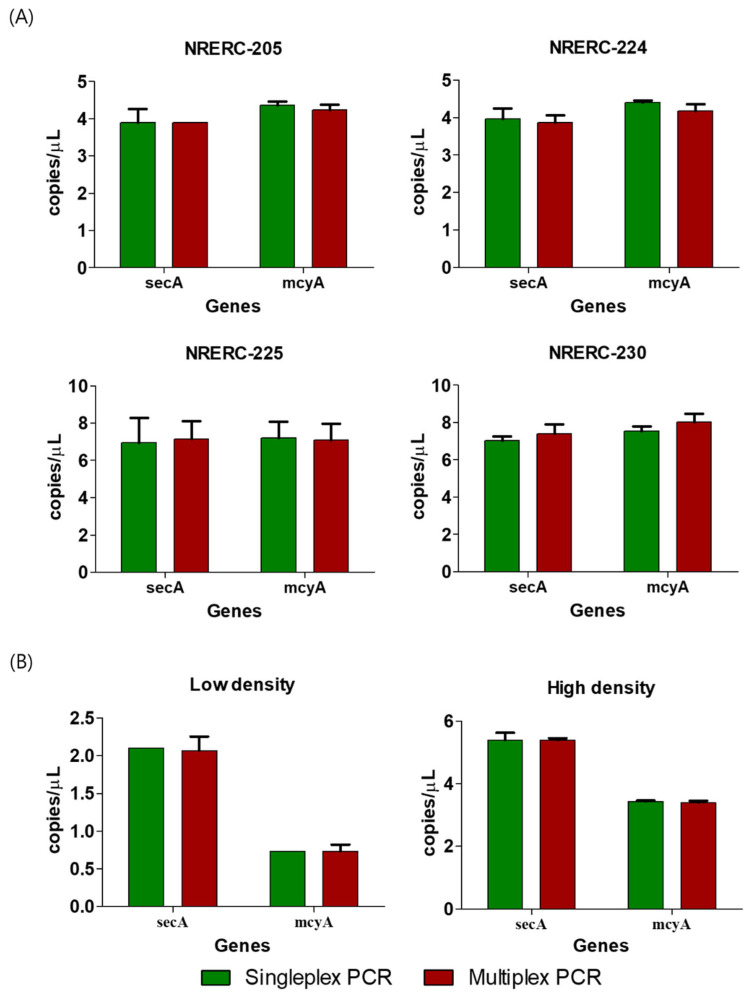
Comparison of sensitivity between *secA* and *mcyA* singleplex PCR and multiplex PCR in ddPCR: (**A**) cultured *Microcystis* strains; (**B**) mixed field populations of *Microcystis* collected from CH site. Data are presented as mean ± SEM. Statistical analysis was performed using an unpaired two-tailed Student’s *t*-test (n = 6). No statistically significant difference was observed (*p* > 0.05).

**Figure 3 toxins-17-00242-f003:**
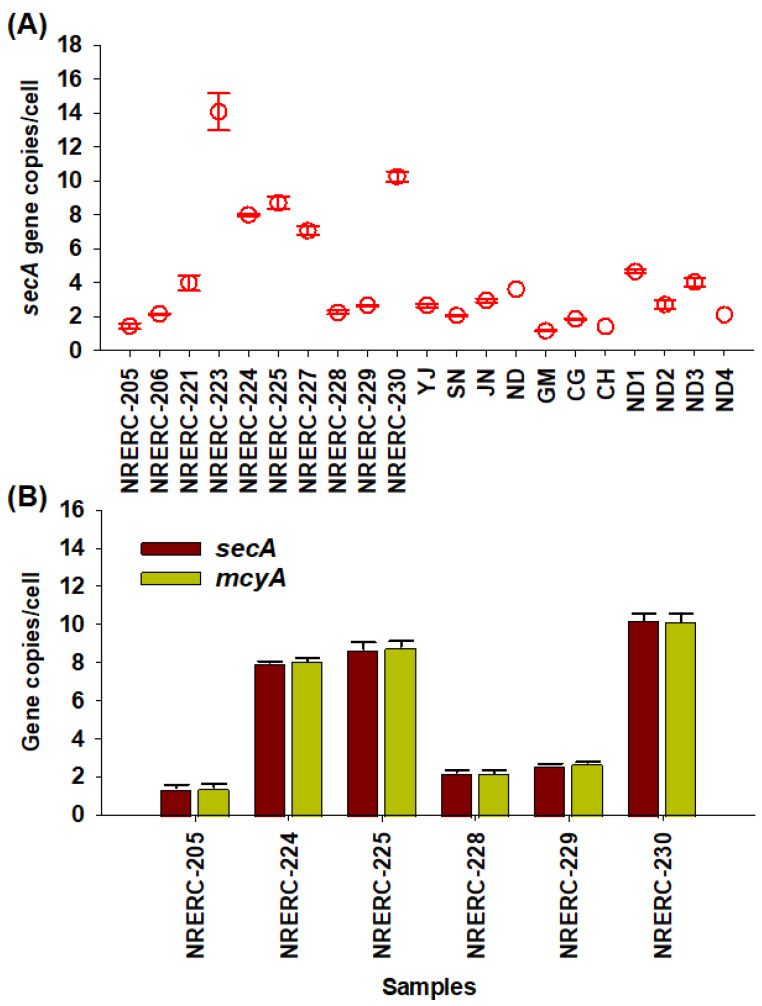
The *secA* gene copy number per cell in *Microcystis* strains and mixed filed populations of *Microcystis* (**A**) and comparison of *secA* and *mcyA* gene copy numbers of toxic *Microcystis* strains (**B**). Data are presented as mean ± SEM (bars above and below the symbols). Statistical analysis was performed using an unpaired two-tailed Student’s *t*-test (n = 6). No statistically significant difference was observed (*p* > 0.05).

**Figure 4 toxins-17-00242-f004:**
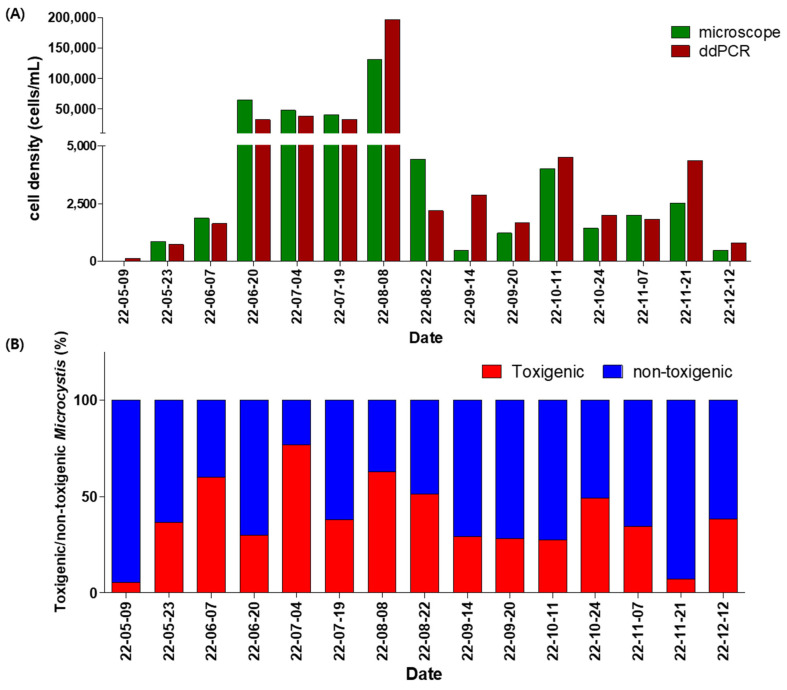
Comparisons of cell densities between microscopic cell counting method and estimation from gene copy number of ddPCR of field samples (**A**), and the ratio of toxigenic/non-toxigenic *Microcystis* biomass (**B**).

**Figure 5 toxins-17-00242-f005:**
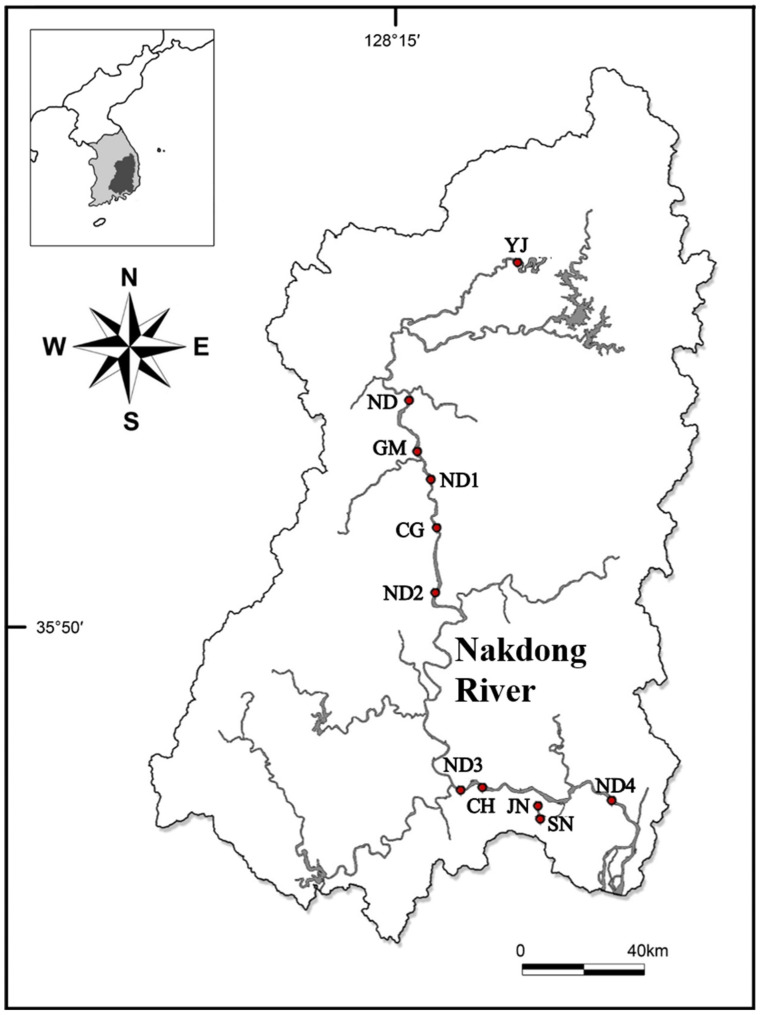
Map of sampling sites in the Nakdong River basin area.

**Table 1 toxins-17-00242-t001:** PCR primers and probes list.

Platform	Target Gene	Name	Sequence(5′ -> 3′)	Reference
ddPCR	*secA*	*secA*-F	GTGTGGGACTAATTCAAGCC	This study
		*secA*-R	GGCCTCATCGATAAGAATGG	
		*secA*-probe	TACACCACTAACAGCGAACTCGGCT	
	*mcyA*	*mcyA*-F	GGTTAGAAGCAGCCGATGG	This study
		*mcyA*-R	GCTCCAAGAACCTCCAGATAAC	
		*mcyA*-probe	TTTTGAATACTTTGCCCCTACGTTTAGA	
	16S rRNA	MICR 184F	GCCGCRAGGTGAAAMCTAA	[56]
		MICR 431R	AATCCAAARACCTTCCTCCC	
		MICR 228F	AAGAGCTTGCGTCTGATTAGCTAGT	
Conventional PCR	*mcyA*	*mcyA*-Cd1F	AAAATTAAAAGCCGTATCAAA	[57]
		*mcyA*-Cd1R	AAAAGTGTTTTATTAGCGGCTCAT	

**Table 2 toxins-17-00242-t002:** Dominant species and sub-dominant species of *Microcystis* in the field samples from May to December in 2022.

Survey Date	Dominant Species (Relative Abundance %)	Sub-Dominant Species (Relative Abundance %)
2022.5.23	*Microcystis* sp. (100)	
2022.6.7	*M. aeruginosa* (46)	*M. viridis* (38)
2022.6.20	*M. aeruginosa* (41)	*M. novacekii* (26)
2022.7.4	*M. aeruginosa* (46)	*M. viridis* (33)
2022.7.19	*M. wesenbergii* (65)	*M. aeruginosa* (23)
2022.8.8	*M. aeruginosa* (61)	*M. novacekii* (12)
2022.8.22	*M. aeruginosa* (70)	*Microcystis* sp. (20)
2022.9.14	*Microcystis* sp. (100)	
2022.9.20	*M. aeruginosa* (56)	*M. ichthyoblabe* (44)
2022.10.11	*M. aeruginosa* (46)	*M. wesenbergii* (31)
2022.10.24	*M. aeruginosa* (83)	*Microcystis* sp. (17)
2022.11.7	*M. wesenbergii* (50)	*Microcystis* sp. (50)
2022.11.21	*Microcystis* sp. (100)	
2022.12.12	*Microcystis* sp. (100)	

**Table 3 toxins-17-00242-t003:** List of cyanobacterial strains used for primer verification, presence of *secA*, *mcyA* genes by conventional PCR, and production of microcystins (MCY) in each strain.

No.	Strain No.	Scientific Name	Isolation Site	*secA* Gene	*mcyA* Gene	MCY
1	NRERC-205	*Microcystis flos-aquae*	Nakdong River	+	+	+
2	NRERC-206	*Microcystis ichthyoblabe*	Nakdong River	+	−	−
3	NRERC-221	*Microcystis aeruginosa*	Youngju Lake	+	−	−
4	NRERC-223	*Microcystis wesenbergii*	Youngju Lake	+	−	−
5	NRERC-224	*Microcystis viridis*	Nakdong River	+	+	+
6	NRERC-225	*Microcystis novacekii*	Banbyeon River	+	+	+
7	NRERC-227	*Microcystis viridis*	Youngsan River	+	−	−
8	NRERC-228	*Microcystis flos-aquae*	Youngsan River	+	+	+
9	NRERC-229	*Microcystis flos-aquae*	Youngsan River	+	+	+
10	NRERC-230	*Microcystis aeruginosa*	Youngsan River	+	+	+
11	NRERC-008	*Aphanizomenon flos-aquae*	Nakdong River	−	−	−
12	NRERC-101	*Dolichospermum planctonicum*	Nakdong River	−	−	−
13	NRERC-450	*Geitlerinema amphibium*	Nakdong River	−	−	−
14	NRERC-501	*Cylindrospermopsis raciborskii*	Nakdong River	−	−	−
15	NRERC-601	*Sphaerospermopsis aphanizomenoides*	Nakdong River	−	−	−
16	NRERC-650	*Cuspidothrix issatchenkoi*	Nakdong River	−	−	−
17	AG10016	*Aphanocapsa* sp.		−	−	−
18	AG20470	*Synechococcus* sp.		−	−	−
19	NIVA-CYA 656	*Dolichospermum flos-aquae*		−	+	+
20	NIVA-CYA 855	*Planktothrix agardhii*		−	+	+

## Data Availability

The original contributions presented in this study are included in the article. Further inquiries can be directed to the corresponding author.

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
