# Peer review of "Molecular Quantification of Total and Toxigenic Microcystis Using Digital-Droplet-Polymerase-Chain-Reaction-Based Multiplex Assay"

_toxins, 2025, doi:10.3390/toxins17050242_

Round 1
Reviewer 1 Report
Comments and Suggestions for Authors
Dear corresponding author:
Below I am including suggestions for you to consider to finalize the manuscript for publication in Toxins.
Summary of Review:
Molecular quantification of total and toxigenic Microcystis us- 2 ing Droplet Digital PCR-based multiplex assay
Title
Ok
Abstract
L15-16. Authors mention that ddPCR detected Microcystis at very low densities. Please add limits of detection data here to inform readers regarding the suitability of the developed analytical method (Section 2.2). Additionally the summary of the results from Nakdong River samples are missing. Even more importantly a mention that the results show that the secA may be a reliable genetic marker for detecting and quantifying Microcystis within mixed cyanobacterial communities.
Key words
Add major cyanobacteria species included in the study.
Introduction
Overall the introduction section focuses on environmental chemistry and monitoring approaches but lack literature review focused on methodology development approaches of target molecules - the main theme of this study. For example, recently popular kit based ELISA is commonly used protocol for the identification of Microcystis metabolites. Additionally review of methods focused on meta-barcoding for the detection of mycotoxin toxicology should be added (https://www.nature.com/articles/s41598-022-14216-8). Regarding PCR based methods additional references reporting limits of detection and quantification of environmental media by comparing Q-PCR and Droplet-PCR https://doi.org/10.1128/AEM.00931-15. Additionally the introduction of the study area Nakdong River is worth including.
Results
L127-130. Explain what does event number and M mean labelled on x-axes.
L146. Mention where is Nakdong River located and the specific sampling site/s.
Discussion
Consider merging discussion with results.
Conclusion
Ok
Materials and Methods
Ok
Supplementary material
Not evaluated
Author contribution
Not evaluated
Acknowledgements
Not evaluated
Author Response
We attached the Respones to comments file.

Reviewer 2 Report
Comments and Suggestions for Authors
Manuscript Title: “Molecular quantification of total and toxigenic Microcystis using Droplet Digital PCR-based multiplex assay"
A summary of the paper.
The study proposes the application of the ddPCR method for monitoring toxigenic and total cyanobacteria of the genus Microcystis during bloom events. The proposal can be a substitute for qPCR because it does not need to produce a standard curve for quantification and obtains results even with samples with a reduced number of cells. Thus, the authors' proposal seems to be promising for ecotoxicological studies of diagnostic analyses in blooms and for ecological risk assessment studies with samples with low cyanobacteria biomass.
Major comments.
The text of the article is of scientific quality. However, some points were not very clear, and I would like the authors to clarify:
Materials and Methods:
- I think that Table 1 would be better located in the Materials and Methods item. I found it a little confusing the way it was in results. Between lines 111 and 119, the text could leave the results and be in Materials and methods. During my reading it was difficult to relate Table 1 (in Results), Figure 1 (in Results) and Table 3 (in Materials and Methods).
- In line 394, Table 1 is cited, but shouldn't it be Table 3?
- Table 3 is confusing. There is no information on whether it was conventional PCR or ddPCR.
- Item 5.1 lines 405-407 require more information on the cyanobacterial culture (for example: inoculum volume, number of cells used initially and at the end of the culture, how cell quantification was monitored).
Results:
- In Fig. 1, indicate in the caption that the original electrophoretic gels are in Supplementary materials;
- Lines 133-139. It was not clear how microcystin-producing species that do not belong to the genus Microcystis did not show positive signals in the ddPCR.
- Lines 237-240: In the results from field samples, it was impossible to determine the number of copies of mcyA, since it was not possible to count the number of toxigenic cells in the field sample. I propose a reflection for the authors: “However, for monitoring, the most important thing is not the cell, but the toxin. So, if the gene is detected in the sample, wouldn't it already be relevant? Even if the number of toxigenic cells is not available.” The tool is excellent, and at this point it would be a way to explore its practicality.
Discussion:
- As mentioned above (item 2), it was not clear why Dolichospermum flos-aquae and Planktothrix agardhii did not show a signal in the ddPCR. In lines 273-275, the authors merely confirmed what they had already said in lines 136-139 of the results. This result needs to be further discussed. The authors may propose hypotheses if they deem them pertinent: a) analyze the fact that they produced primers and probes from non-homologous regions of the gene for the species used; b) it is possible that some species may produce microcystin without the mcyA gene; and c) that the ELISA result for D. flos-aquae and P. agardhii may have been a false positive.
Follow with minor comments :
- Tables must be formatted according to the journal's standards
- Use SI units to show copies/µL to 106 copies/L
- Line 166 – Adjust the word “silmilar” to “similar”
- Line 283 and 293 - use an article before DdPCR: The ddPCR
Author Response
We attatched the Responses to Comments file.
